# Requirement for p62 acetylation in the aggregation of ubiquitylated proteins under nutrient stress

Zhiyuan You [1], Wen-Xue Jiang[2], Ling-Yun Qin[2], Zhou Gong[2], Wei Wan[1], Jin Li[1], Yusha Wang[1], Hongtao Zhang[1], Chao Peng [3], Tianhua Zhou[1], Chun Tang [2]* & Wei Liu [1,4]*

Autophagy receptor p62/SQSTM1 promotes the assembly and removal of ubiquitylated proteins by forming p62 bodies and mediating their encapsulation in autophagosomes. Here we show that under nutrient-deficient conditions, cellular p62 specifically undergoes acetylation, which is required for the formation and subsequent autophagic clearance of p62 bodies. We identify K420 and K435 in the UBA domain as the main acetylation sites, and TIP60 and HDAC6 as the acetyltransferase and deacetylase. Mechanically, acetylation at both K420 and K435 sites enhances p62 binding to ubiquitin by disrupting UBA dimerization, while K435 acetylation also directly increases the UBA-ubiquitin affinity. Furthermore, we show that acetylation of p62 facilitates polyubiquitin chain-induced p62 phase separation. Our results suggest an essential role of p62 acetylation in the selective degradation of ubiquitylated proteins in cells under nutrient stress, by specifically regulating the assembly of p62 bodies.

[1] Department of Biochemistry and Department of Cardiology of the Second Affiliated Hospital, Zhejiang University School of Medicine, Hangzhou 310058, China. [2] CAS Key Laboratory of Magnetic Resonance in Biological Systems, State Key Laboratory of Magnetic Resonance and Atomic Molecular Physics, National Center for Magnetic Resonance at Wuhan, Wuhan Institute of Physics and Mathematics of the Chinese Academy of Sciences, Wuhan 430071, China. [3] National Center for Protein Science Shanghai, Institute of Biochemistry and Cell Biology, Shanghai Institutes of Biological Sciences, Chinese Academy of Sciences, Shanghai 200031, China. [4] Joint Institute of Genetics and Genomics Medicine between Zhejiang University and University of Toronto, Hangzhou 310058, China. *email: tanglab@wipm.ac.cn; liuwei666@zju.edu.cn

Cells use various mechanisms to maintain homeostasis, which is essential for cell survival, metabolism and growth. The ubiquitin-proteasome system (UPS) and autophagy are two major quality control pathways through which cells remove misfolded/unfolded proteins[1–3]. While UPS recognizes and degrades polyubiquitylated (poly-Ub) misfolded proteins through 26S proteasomes, autophagy eliminates poly-Ub protein aggregates by encapsulating them into autophagosomes[1–3]. Under different stress conditions, excessive level of misfolded proteins caused by overproduction and/or UPS damage in cells can lead to the formation of large amounts of protein aggregates[4–6]. Deficiency in autophagic clearance of these protein aggregates is associated with the development of many aging-related diseases including cancer and neurodegenerative diseases[7–9].

Autophagosomes capture protein aggregates through a panel of cargo receptor proteins. These cargo receptors use their ubiquitin-associated (UBA) domain to bind to ubiquitylated proteins, and their LC3-interaction region (LIR) to bind to Atg8/LC3 on autophagic membranes[10–15]. p62, the first autophagic cargo receptor identified in mammalian cells, plays a key role in mediating the formation and autophagic clearance of intracellular protein aggregates[13,16,17]. However, biochemical and structural analyses have shown that p62 possesses a low natural affinity for ubiquitin, while the dimerization between their UBA domains prevents them from binding ubiquitin[18,19]. Therefore, under stress conditions, in order to assemble ubiquitylated proteins, p62 needs post-translational modifications such as phosphorylation and/or ubiquitylation to disrupt the UBA dimerization, in addition to self-oligomerization through its PB1 domain[20–22]. Physiologically, autophagic clearance of these p62-positive aggregates ensures not only that misfolded proteins are removed, but also that the homeostatic level of p62 itself is preserved. This is important because the accumulation of intracellular p62 is associated with the malignant transformation of autophagy-deficient cells[9,17]. Interestingly, it was recently suggested that like other intracellular membraneless compartments, the assembled p62-positive protein aggregates are actually liquid droplets formed by phase separation, which depends on the interaction

between p62 and ubiquitin and can be regulated by post-translational modification of p62[23,24]. These cytoplasmic aggregates assembled from p62 and poly-Ub proteins are also called p62 inclusion bodies or p62 bodies.

In cells under nutrient-deficient conditions, cytoplasmic components are usually captured by autophagosomes in a non-selective manner, and cargo degradation serves to supplement energy and amino acids for cell survival[25,26]. p62 bodies have long been observed in such cells at the early stage of starvation[13,27]. They localize to autophagosome formation sites and assemble via self-oligomerization, but this process does not rely on LC3 and ULK1[27], which suggests that p62 body formation is independent of autophagosomes biogenesis. Interestingly, in these cells, neither phosphorylation nor ubiquitylation of p62 was detected[20,22]. Starvation-induced autophagy is generally considered to be non-selective, and the degradation of intracellular p62 has been used to determine autophagy flux; however, the regulation of p62 body formation and its physiological significance in starving cells have not been elucidated.

In this study, we report that in nutrient-deficient cells, the formation of p62 bodies is specifically regulated by acetylation modification of p62, which is mediated by activation of the acetyltransferase TIP60. We demonstrate that acetylation markedly enhances the affinity of p62 for ubiquitin, which facilitates p62 phase separation and the clearance of ubiquitylated proteins. We further show that the formation of p62 bodies, in addition to promoting their own degradation by selective autophagy, improves the survival of cells under nutrient starvation conditions.

## Results

**Acetylation of p62 induced by nutrient deficiency.** Previous studies have identified that acetylation plays important roles in autophagy regulation, and a number of autophagy related proteins (ATGs) undergo acetylation/deacetylation during cell starvation[28–32]. To test whether p62 is an acetylated protein, we analyzed the acetylation of exogenously expressed p62 (Fig. 1a)

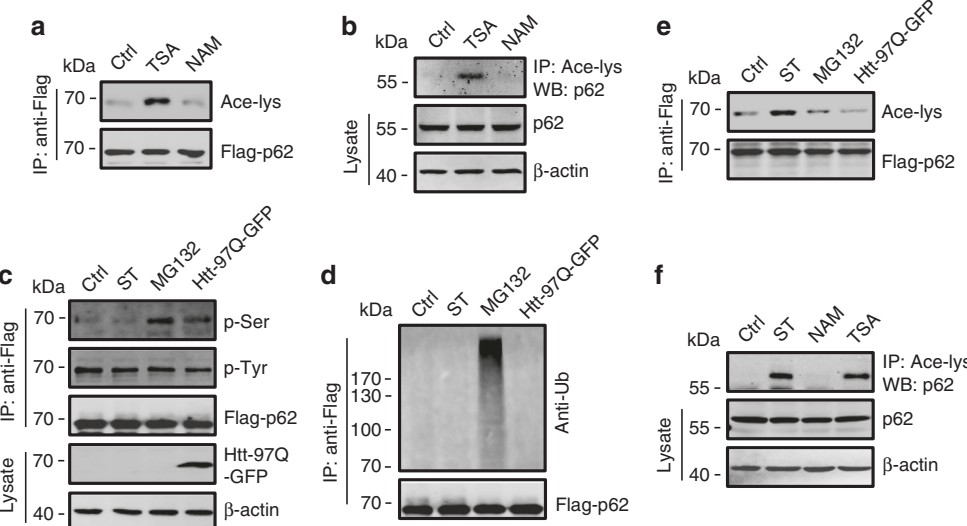

**Fig. 1 Starvation induces acetylation of p62. a** Acetylation of exogenous Flag-p62 in HEK293 cells treated with deacetylase inhibitors TSA or NAM. Flag-p62 was immunoprecipitated with anti-Flag, and the precipitates were analyzed using an anti-acetyl-lys antibody (Ace-lys). **b** Acetylation of endogenous p62 in HEK293 cells treated with TSA or NAM. p62 acetylation was analyzed by immunoprecipitation with an anti-acetyl-lys antibody followed by western blotting for p62. **c–e** Phosphorylation (**c**), ubiquitylation (**d**), and acetylation (**e**) of Flag-p62 in HEK293 cells incubated in starvation medium (ST), treated with proteasome inhibitor MG132, or overexpressing Htt-poly97Q-GFP. Flag-p62 was immunoprecipitated with anti-Flag and analyzed by western blot using anti-phospho-serine (**c**), anti-phospho-tyrosine (**c**), anti-ubiquitin (anti-Ub) (**d**), and anti-acetyl-lys (**e**) antibodies, respectively. **f** Cell starvation induces the acetylation of endogenous p62 in HEK293 cells. Source data are provided as Source Data file.

and endogenous p62 (Fig. 1b) in cells treated with trichostatin A (TSA), a broad spectrum inhibitor of HDAC family deacetylases, and nicotinamide (NAM), an inhibitor of SIRT family deacetylases. Using a specific antibody against acetylated lysine, we detected strong acetylations of both exogenous and endogenous p62 in TSA-treated but not NAM-treated cells (Fig. 1a, b). We then examined p62 acetylation in starved cells, and in cells exposed to two known stressors that cause p62 phosphorylation and/or ubiquitylation, i.e., treatment with the proteasome inhibitor MG132 and overexpression of aggregate-prone huntingtin protein (Htt-poly-97Q)[20–22]. Using antibodies against phospho-serine, phospho-tyrosine or ubiquitin, we found that unlike MG132 or Htt-poly-97Q, cell starvation induced neither the

phosphorylation nor ubiquitylation of p62 protein (Fig. 1c, d) but, surprisingly, it stimulated acetylation of exogenous and endogenous p62 (Fig. 1e, f). These results suggest that p62 is an acetylated protein and the acetylation can be specifically triggered by cell starvation.

**Interaction of p62 and histone acetyltransferase TIP60.** To identify the acetyltransferase of p62, we first examined the potential interaction between p62 and each of the most common cell metabolism-related acetyltransferases. Co-immunoprecipitation analysis identified a strong and specific association between p62 and acetyltransferase TIP60 in HEK293T cells (Fig. 2a, b). p62 is

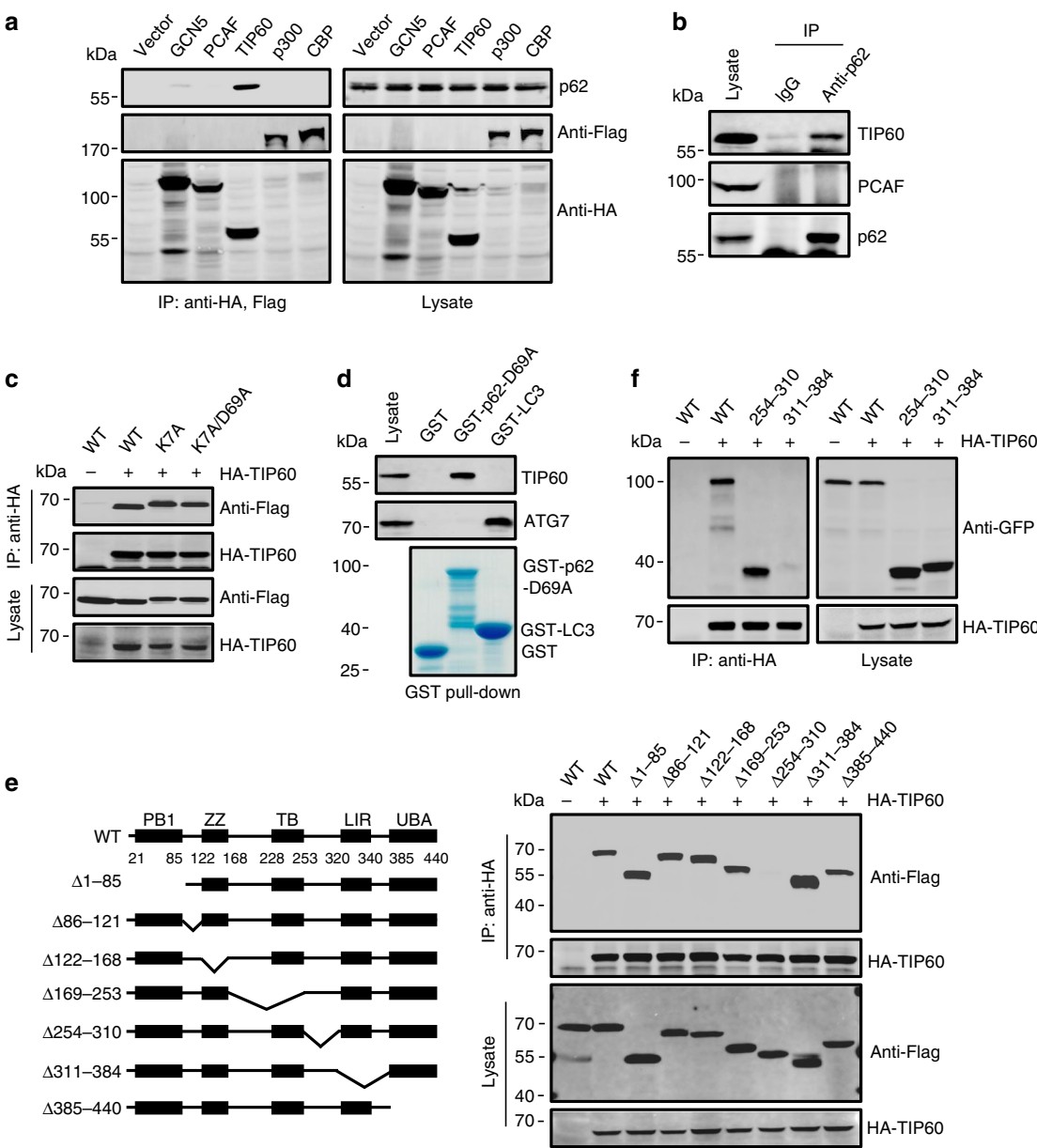

**Fig. 2 Interaction of p62 and TIP60. a** Co-immunoprecipitation of endogenous p62 with exogenously expressed acetyltransferases from HEK293T cells. Acetyltransferases were immunoprecipitated by anti-HA or anti-Flag and the precipitates were analyzed using anti-p62. **b** Co-precipitation of endogenous TIP60 with p62 from HEK293T cells. **c** Co-immunoprecipitation of p62 mutants with HA-TIP60 in HEK293T cells. HA-TIP60 was immunoprecipitated by anti-HA and the precipitates were analyzed using anti-Flag. **d** HEK293T cell lysates were incubated with purified GST-p62-D69A or GST-LC3B, and bound TIP60 and ATG7 were detected by western blot using anti-TIP60 and anti-ATG7. **e, f** Co-precipitation of Flag-tagged (**e**) or GFP-tagged (**f**) p62 truncated mutants with HA-TIP60 from HEK293T cells. HA-TIP60 precipitates were analyzed using anti-Flag and anti-GFP, respectively. Source data are provided as Source Data file.

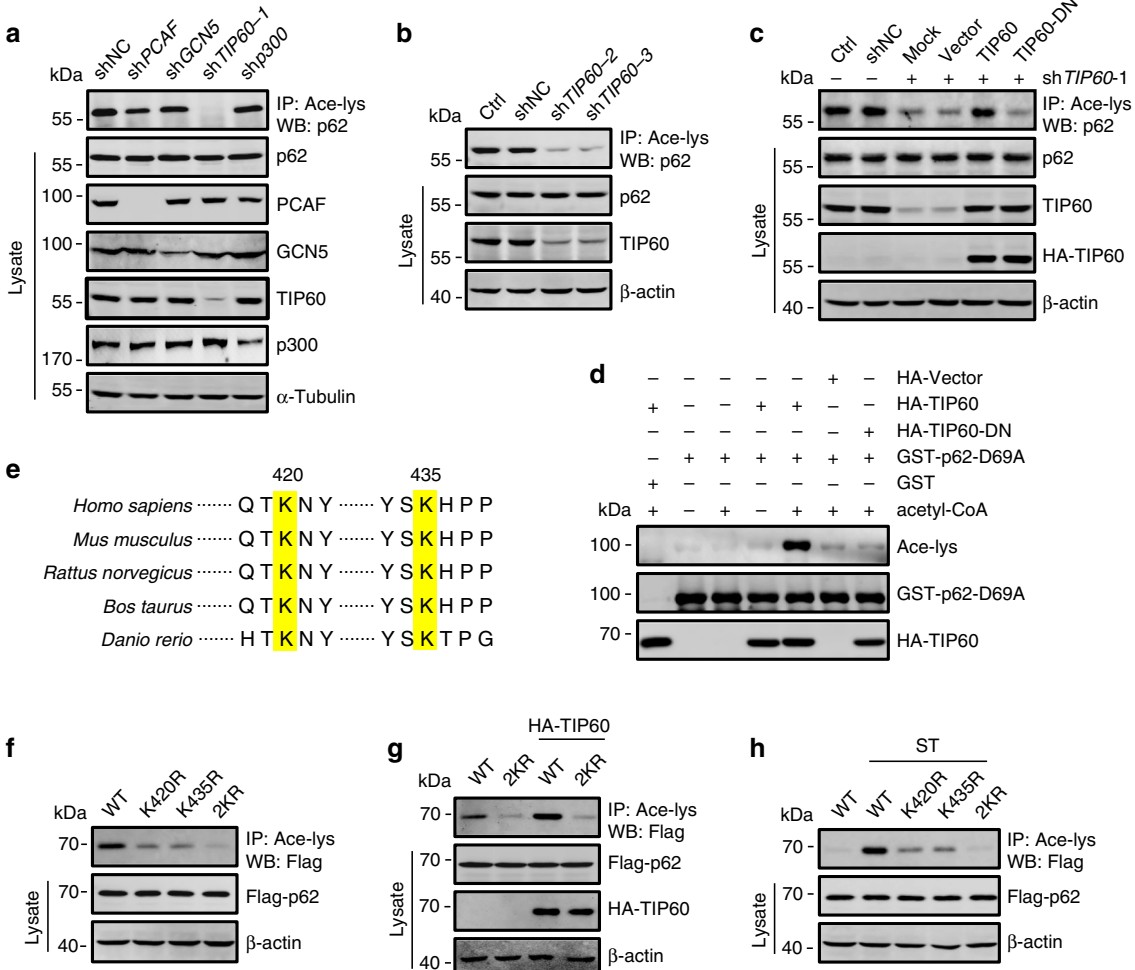

**Fig. 3 TIP60 acetylates p62 at K420 and K435. a–c** Acetylation of p62 in HeLa cells infected with lentivirus expressing each of the indicated acetyltransferase shRNAs (**a**, **b**) and with HA-TIP60 or HA-TIP60-DN transfection 48 h after *TIP60* shRNA infection (**c**). **d** In vitro p62 acetylation assay. Purified GST-p62-D69A from *E. coli* was incubated with HA-TIP60 or HA-TIP60-DN immunoprecipitated from HEK293T cells, in the presence or absence of acetyl-CoA. p62 acetylation was analyzed by western blot using anti-acety-lys. **e** Alignment of p62 amino acid sequence from various species. Yellow shading indicates the conserved K420 and K435. **f–h** Acetylation of p62 mutants expressed in HeLa cells (**f**, **g**) or HEK293 cells (**h**). The cells were transfected with or without HA-TIP60 (**g**), or treated with or without starvation (**h**). 2KR, both Lys420 and Lys435 residues were replaced by Arg. Source data are provided as Source Data file.

very prone to form multimers through its PB1 domain[33–35]; therefore, to exclude the possibility that the co-precipitation results were due to p62 multimerization, we checked the interaction of TIP60 with the multimerization-impaired mutants p62-K7A and p62-K7A/D69A in which the interaction surface of the PB1 domain is compromised[33,34]. We found that p62-K7A and p62-K7A/D69A were co-precipitated with TIP60 to the same extent as wild type (WT) p62 (Fig. 2c). Further, in vitro GST pull-down assays indicated that purified recombinant GST-p62-D69A specifically pulled down TIP60 from the cell lysates, whereas purified GST-LC3B only pulled down its known interaction partner ATG7 (Fig. 2d).

To further verify the interaction between p62 and TIP60, and map the responsible region in p62, we created different truncated forms of p62 (Fig. 2e) and examined their co-precipitation with TIP60. HA-TIP60 failed to co-precipitate mutant p62 with a deletion of amino acids 254–310, which are located between the TRAF6-binding (TB) domain and the LIR domain (Fig. 2e). In addition, when expressed in cells, a GFP-tagged peptide containing only amino acids 254–310 of p62 co-immunoprecipitated with TIP60 (Fig. 2f).

Taken together, these results suggest a direct and specific interaction between p62 and acetyltransferase TIP60.

**p62 is acetylated by TIP60 at Lysine 420 and Lysine 435**. We tested whether p62 is an acetylation substrate of TIP60. Knockdown in HeLa cells of *TIP60* but not the other acetyl-transferase genes largely reduced the basal acetylation level of p62 (Fig. 3a). The reduction could also be induced by two other *TIP60* shRNAs (Fig. 3b) and retrieved by re-introduction of RNAi-resistant WT TIP60 but not the acetyltransferase-deficient TIP60 (TIP60-DN)[36] into the knockdown cells (Fig. 3c). We then carried out in vitro acetylation assays by incubating recombinant GST-p62-D69A purified from *E. coli* with HA-TIP60 immunoprecipitated from HEK293T cells. In the presence of acetyl-coenzyme A (acetyl-CoA), a strong acetylation of GST-p62-D69A was observed (Fig. 3d), indicating that TIP60 directly acetylates p62.

Mass spectrometry analysis of p62 from the in vitro acetylation reaction suggested two potential acetylation sites, K420 and K435 (Supplementary Fig. 1a), both of which are located in UBA domain of p62 (p62-UBA) and highly conserved among species (Fig. 3e). These two lysines were also suggested by mass spectrometry analysis of Flag-p62 from TSA-treated HEK293T cells (Supplementary Fig. 1b), which implies that they are the main acetylation sites on p62. To verify this, Flag-tagged p62 mutants in which each

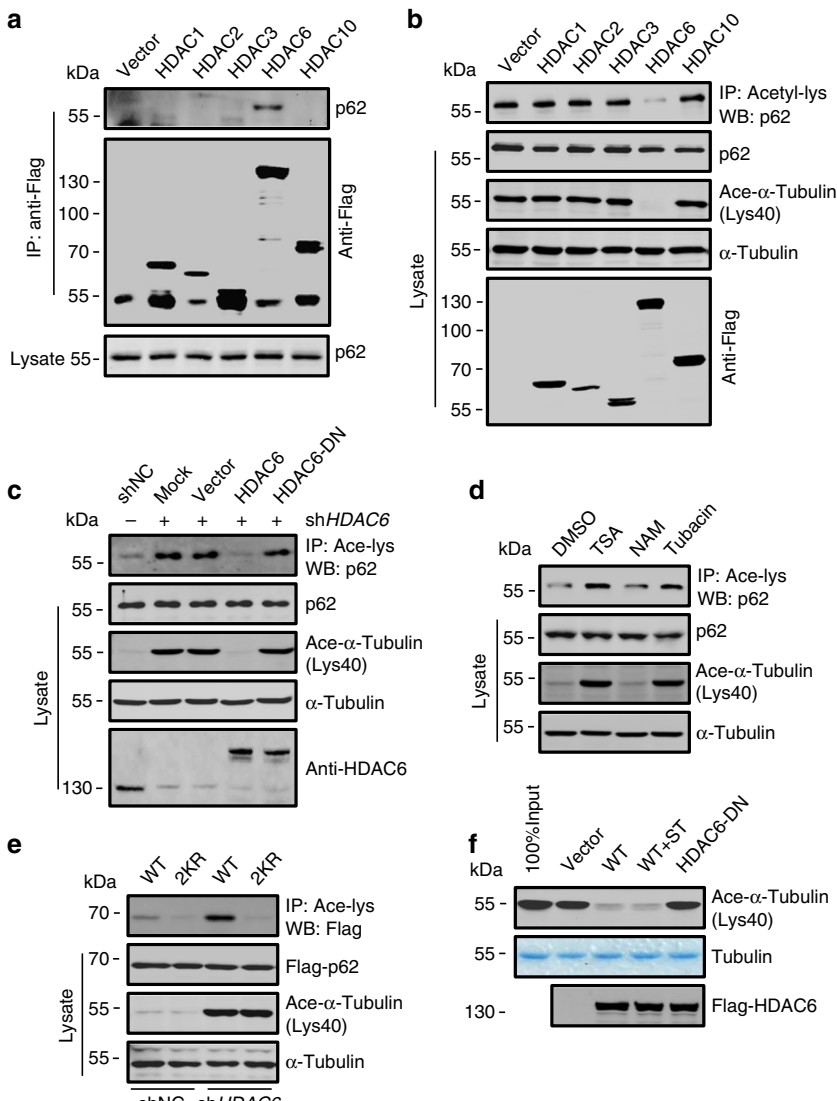

**Fig. 4 p62 is deacetylated by HDAC6. a** Co-precipitation of endogenous p62 with each of the indicated Flag-tagged deacetylases from HEK293T cells. **b** p62 acetylation in HeLa cells overexpressing the individual deacetylases. **c** p62 acetylation in HEK293 cells transfected with Flag-tagged WT HDAC6 or the HDAC6-DN 48 h after *HDAC6* shRNA infection. **d** Acetylation of p62 in HEK293 cells treated with TSA, NAM or Tubacin. **e** Acetylation of Flag-tagged WT p62 and p62-2KR in HEK293 cells infected with lentivirus expressing *HDAC6* shRNA. **f** Purified assembled microtubules were incubated with Flag-tagged WT HDAC6 or HDAC6-DN immunoprecipitated from fed or starved HEK293T cells. Acetylation of α-tubulin in the incubation was detected by western blot using an antibody against acetylated α-tubulin (Lys40). Source data are provided as a Source Data file.

of the two lysine residues was changed to arginine via site-directed mutagenesis, were transfected into HeLa cells. Compared with WT p62, p62-K420R and p62-K435R showed markedly reduced acetylation, and the double-mutant p62-K420R/K435R (p62-2KR) was hardly acetylated at all (Fig. 3f). In addition, the acetylation of these mutants was much weaker than WT p62 in cells overexpressing TIP60 or subjected to starvation (Fig. 3g, h). Taken together, these data suggest that K420 and K435 are the main sites of TIP60 acetylation in p62.

**p62 is deacetylated by HDAC6.** The stimulation of p62 acetylation by TSA rather than NAM suggests the involvement of HDAC family deacetylases. Using a strategy similar to the identification of p62 acetyltransferase, we detected the specific interaction between p62 and HDAC6 (Fig. 4a), which supports the finding of a previous study[37]. In addition, overexpression in cells of HDAC6 rather than other HDAC members greatly reduced p62 acetylation (Fig. 4b). Accordingly, knockdown of *HDAC6*

increased p62 acetylation and the increase was abolished by re-expression of RNAi-resistant WT HDAC6 but not the deacetylase-dead HDAC6 (HDAC6-DN)[38] (Fig. 4c). Furthermore, p62 acetylation was stimulated by the specific HDAC6 inhibitor Tubacin (Fig. 4d), and knockdown of *HDAC6* failed to raise the acetylation level of p62-2KR (Fig. 4e). Taken together, these data suggest that HDAC6 is a deacetylase of p62 which targets K420 and K435.

Activation of TIP60 has previously been observed in cells under starvation[30]. To determine whether inactivation of HDAC6 is also involved in starvation-stimulated p62 acetylation, purified porcine brain-derived microtubules were incubated with HDAC6 immunoprecipitated from fed or starved cells. We found that HDAC6 from starved cells showed the same deacetylation activity as HDAC6 from fed cells on acetylated α-tubulin (Lys40) (Fig. 4f), which suggests that HDAC6 was not inhibited in starved cells. Therefore, p62 acetylation stimulated by cell starvation may mainly result from the activation of TIP60.

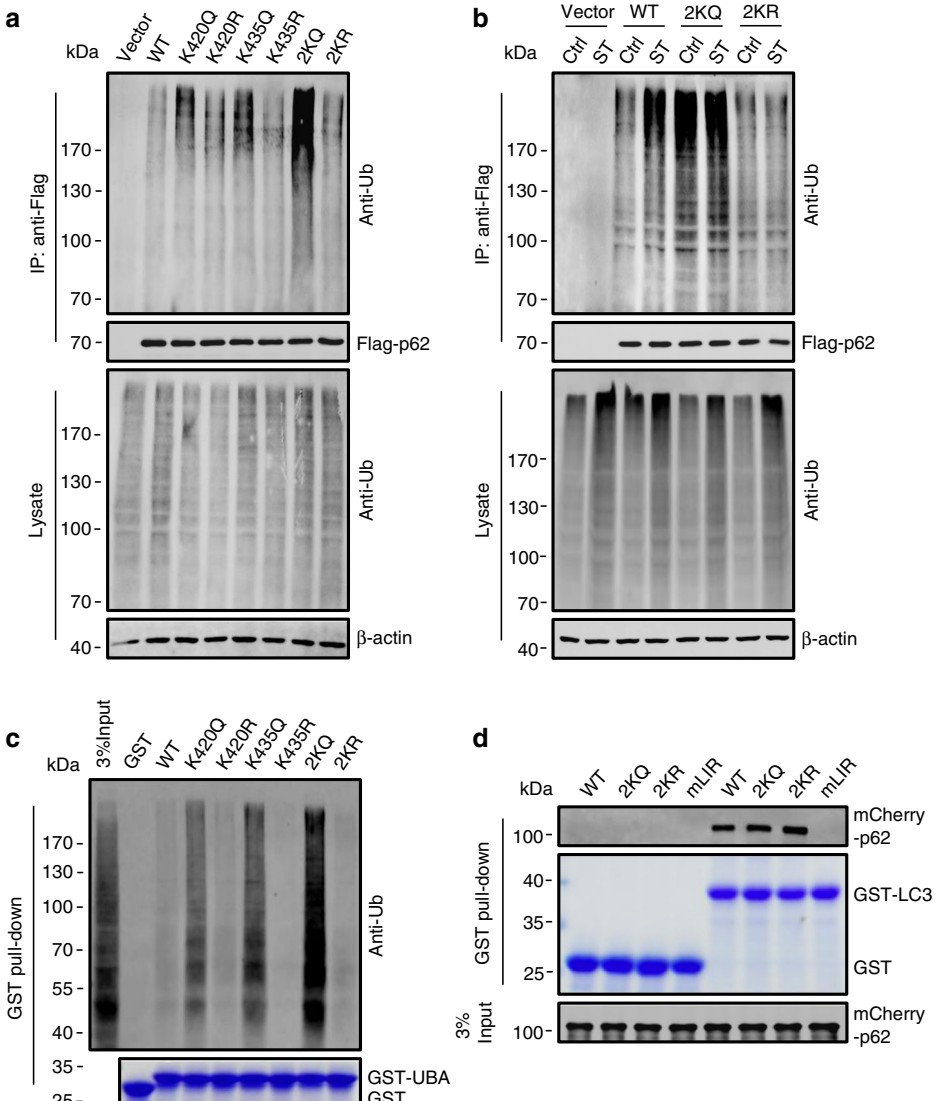

**Fig. 5 Acetylation promotes p62-ubiquitin binding. a, b** Co-precipitation of ubiquitylated proteins with Flag-p62. Flag-p62 was immunoprecipitated by anti-Flag from *p62*-KO HEK293 cells transiently expressing Flag-tagged WT p62 or each of the p62 mutants and co-precipitated ubiquitylated proteins were detected using anti-Ub. 2KQ, both Lys420 and Lys435 residues were replaced by Gln. **c** GST pull-down assay of the binding of p62-UBA and ubiquitylated proteins. Cell lysates of MG132-treated HeLa cells were incubated with purified GST-tagged UBA of WT p62 or each of the indicated p62 mutants. The GST-UBA-bound ubiquitylated proteins were detected using anti-Ub. **d** GST pull-down assay showing the interaction of purified GST-LC3B and purified mCherry-tagged WT or mutant p62 proteins. mLIR: p62 with LIR mutation. Source data are provided as a Source Data file.

**Acetylation promotes the binding of p62 to ubiquitin**. The localization of K420 and K435 in the UBA domain prompted us to speculate that the acetylation at these sites may affect p62-ubiquitin binding. We constructed the acetylation-mimetic mutants p62-K420Q, p62-K435Q, and p62-K420Q/K435Q (p62-2KQ) and examined their association with ubiquitylated proteins in *p62*-KO cells. Compared with WT p62, p62-K420Q and p62-K435Q co-precipitated more ubiquitylated proteins, and p62-2KQ displayed the strongest co-precipitation capacity (Fig. 5a). To exclude the possibility that the ubiquitin signals are derived from the ubiquitylation of p62, we immunoprecipitated p62 proteins from cells using a urea-rich buffer which abolishes non-covalent bonding, then stained them with anti-ubiquitin. We found that compared with p62 ubiquitylation stimulated by the proteasome inhibitor MG132, neither WT p62 nor the p62 mutants exhibited detectable ubiquitylation in cells (Supplementary Fig. 2a). We then checked the binding of p62 to ubiquitylated proteins in starved cells. Cell starvation strongly promoted the binding of WT p62 but not p62-

2KR, and did not alter the binding of p62-2KQ, to intracellular ubiquitylated proteins (Fig. 5b and Supplementary Fig. 2b). Further, we performed in vitro GST pull-down assays by incubating purified recombinant p62-UBA with cell lysates from MG132-treated HeLa cells. The acetylation-mimetic GST-p62-UBAs were much more effective at pulling down ubiquitylated proteins than WT and non-acetylation-mimetic GST-p62-UBAs (Fig. 5c). Finally, we tested whether p62 acetylation affects the direct interaction between p62 and LC3. Recombinant acetylation- and non-acetylation-mimetic p62 proteins were purified and incubated with purified GST-LC3B protein. In vitro GST pull-down assays indicated that acetylation of p62 had no effect on p62-LC3B direct binding (Fig. 5d). Taken together, these data suggest that acetylation at K420 and K435 significantly enhance the binding of p62 to ubiquitin.

**Acetylation at K420 and K435 disrupts UBA dimerization.** To uncover the mechanism of how acetylation modulates the interaction between p62 and ubiquitin, we resorted to nuclear

magnetic resonance (NMR) spectroscopy. Owing to a slow exchange timescale with respect to the chemical shift differences, the NMR peaks for the dimeric form of p62-UBA were distinct from those for the monomeric form of p62-UBA (Supplementary

Fig. 3a), as shown previously[18]. The $^1$H-$^{15}$N heteronuclear single quantum coherence (HSQC) NMR spectra showed that the WT p62-UBA is predominantly dimeric at the concentration range between 20 and 200 μM (Fig. 6a). Based on the peak volumes for

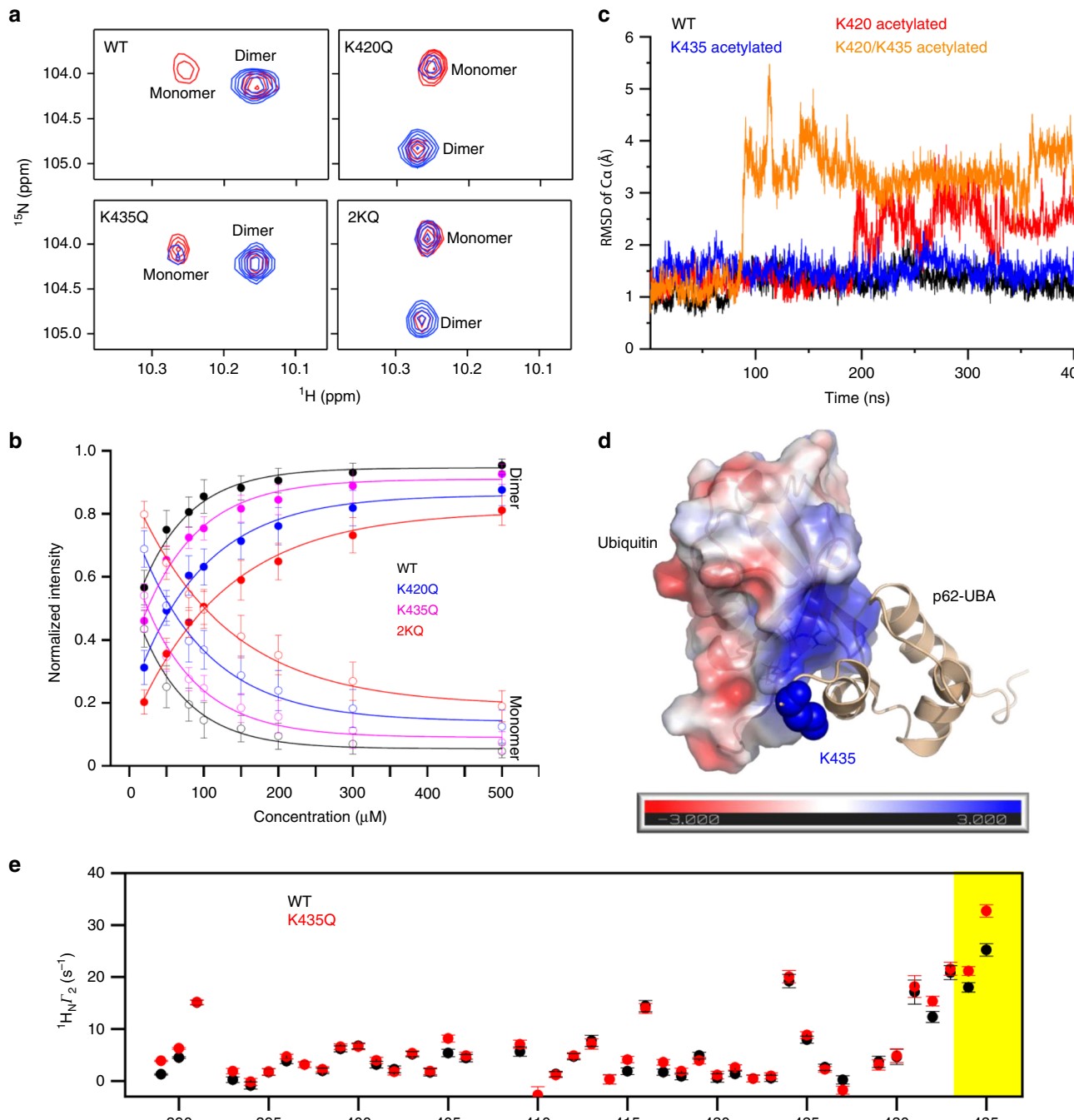

**Fig. 6 Acetylation inhibits UBA dimerization and directly increases the affinity of p62 for ubiquitin. a** The W12 $H_{\varepsilon1}$-$N_{\varepsilon1}$ region of the $^1$H-$^{15}$N HSQC spectra of $^{15}$N-labeled WT and acetylation-mimicking mutants of p62-UBA were measured at high (200 μM, blue) and low (20 μM, red) concentrations. **b** The dissociation constant of each of the indicated p62-UBA proteins was calculated based on the relative peak volumes for dimeric and monomeric species at different concentrations. The residues used for calculation are S399, L402, G405, S407, G411, W412, T419, and G425 $H_{\varepsilon1}$-$N_{\varepsilon1}$; the error bars indicate 1S.D. in the averaged relative peak volume at each protein concentration. **c** Representative trajectories of MD simulation for WT, K420 acetylated, K435 acetylated and K420/K435 acetylated p62-UBA dimers. Fluctuations in Cα RMSD of the dimer are plotted. **d** Structural model of the complex of p62-UBA and ubiquitin. The ubiquitin molecule is colored according to electrostatic potential, which is displayed on a scale from red (−3 kT/e) to blue (+3 kT/e). The side chain of K435 in p62-UBA is shown as blue spheres. **e** Paramagnetic relaxation enhancement for the transverse relaxation rate $\Gamma_2$ of backbone amide protons for $^{15}$N-labeled WT or K435Q mutant of p62-UBA in complex with isotopically unlabeled ubiquitin carrying a nitroxide spin radical tag at the E24C site. The error bars indicate 1S.D. in the fitting of $\Gamma_2$ rates. The yellow shaded area indicates the unstructured C-terminal tail of p62. Source data are provided as a Source Data file.

**Table 1 Equilibrium constant of p62-UBA dimer and p62-UBA-ubiquintin.**

| Proteins | $K_{D,dimer}$ (µM) | $\Delta G$ (kJ per M) | $K_{D,complex}$ (µM) |
|---|---|---|---|
| WT | 6.8 ± 3.7 | −29.5 ± 1.3 | 40.0 ± 2.0 |
| K420Q | 41.2 ± 14.0 | −25.0 ± 0.8 | 34.3 ± 2.4 |
| K435Q | 15.5 ± 5.6 | −27.4 ± 0.9 | 30.4 ± 0.7 |
| 2KQ | 106.3 ± 19.0 | −22.7 ± 0.4 | 33.8 ± 3.7 |

The dissociation constants $K_D$ for p62-UBA dimer interactions and the p62-UBA-ubiquitin interactions were calculated from Fig. 6b and Supplementary Fig. 4b, respectively. The stability of p62-UBA dimer in terms of free energy change is also tabulated, with the errors propagated from the uncertainties of $K_{D,dimer}$

monomeric and dimeric species of p62-UBA at different concentrations, we determined that the $K_{D,dimer}$ value of WT p62-UBA dimerization is ~6.8 µM, which is consistent with the previous reports[18,39] (Fig. 6b, Table 1 and Supplementary Fig. 3a). Analysis of the K420Q, K435Q, and 2KQ mutants of p62-UBA revealed an increased population of the monomer species, which was particularly marked for the p62-UBA-2KQ mutant (Fig. 6b, Table 1 and Supplementary Fig. 3a). We also determined the $K_{D,dimer}$ values for these three mutants. The K420Q and K435Q mutations destabilize the p62-UBA dimer by about 6-fold and 2-fold, respectively, whereas the mutations at both sites are largely additive in terms of free energy change and destabilize the dimer by about 17-fold (Fig. 6b, Table 1 and Supplementary Fig. 3a).

Using molecular dynamics (MD) simulations, we then assessed how real acetylations at the K420 and K435 sites impact the stability of p62-UBA dimers. Single acetylation at K420 and acetylations at both sites increased the dynamic fluctuations of backbone heavy atoms during the MD simulations from about 1 Å to more than 3 Å (Fig. 6c). Consistent with the experiment with acetylation-mimicking mutants, acetylation at K435 only slightly increased the root-mean-square deviations (RMSD) of backbone heavy atoms during the simulations (Fig. 6c). As a control, we analyzed the unmodified and modified p62-UBA monomers, and found that their backbone RMSD values during the MD simulations were essentially the same (Supplementary Fig. 3b). Therefore, the increase in backbone RMSD upon acetylation is mainly caused by the destabilization of the dimer.

**K435 acetylation directly increases UBA-ubiquitin binding.** Ubiquitin monomers only interact with the monomeric species of p62-UBA[18,39]; therefore, ubiquitin binding to p62-UBA is coupled to the dissociation equilibrium of the p62-UBA dimer. We collected a series of HSQC NMR spectra for the $^{15}$N-labeled WT p62-UBA protein and acetylation-mimicking p62-UBA mutant proteins, with the addition of increasing concentrations of unlabeled ubiquitin protein (Supplementary Fig. 4a). Ubiquitin titration caused chemical shift perturbations (CSPs) to a subset of peaks associated with the monomeric species of p62-UBA. Fitting the CSPs as a function of ubiquitin concentration to a one-site binding isotherm, and considering the available p62-UBA monomer concentration, we were able to determine the $K_{D,complex}$ values for the interaction between ubiquitin and p62-UBA. The $K_{D,complex}$ determined for the WT p62-UBA was ~40 µM, whereas the $K_{D,complex}$ values for the mutants were all lower, especially for K435Q (Table 1 and Supplementary Fig. 4b). This suggests that K435 may be involved in binding with ubiquitin.

K435 is located near in the C-terminal tail of p62-UBA, which is missing in the crystal structures of p62-UBA due to its high flexibility[18]. Nevertheless, we were able to model the C-terminal tail in the protein complex, which showed that K435 is near a positively charged patch on ubiquitin (Fig. 6d). Therefore, acetylation of K435 or the K435Q mutation may alleviate the

unfavorable electrostatic repulsion between the two proteins. To validate this, we introduced a nitroxide paramagnetic probe at the E24C site of ubiquitin[40], and measured the intermolecular paramagnetic relaxation enhancement (PRE) values for the $^{15}$N-labeled p62-UBA monomer, either the WT or the K435Q mutant. Compared to the WT p62-UBA in complex with the paramagnetically tagged ubiquitin, the K435Q mutant of p62-UBA exhibited increased PRE values for the C-terminal residues (Fig. 6e). This means that the C-terminal tail of p62-UBA is more tightly packed against ubiquitin upon modification or mutation at K435, which accounts for the higher affinity.

**Acetylation enhances p62 body formation.** Acetylation of p62 increases the binding of p62 to ubiquitylated proteins, which suggests a regulatory role of the acetylation on the formation of p62 bodies in cells. To check this, we generated *p62*-KO HEK293 cells stably expressing WT p62, p62-2KQ or p62-2KR. WT p62 formed a few puncta in fed cells, and this number was increased by starvation (Fig. 7a, b). In comparison, p62-2KQ formed numerous punctate structures even under nutrient-rich conditions, and punctum formation was not further stimulated by cell starvation (Fig. 7a, b). Accordingly, p62-2KR formed almost no puncta in cells under both conditions (Fig. 7a, b). These results suggest an essential role of p62 acetylation in starvation-induced p62 body formation. We then examined the effect of p62 acetylation on p62 phase separation as described previously[23,24]. Purified recombinant mCherry-tagged WT p62, p62-2KQ, or p62-2KR was incubated with purified linear polyubiquitin chains (8 × Ub). Polyubiquitin chains clearly induced phase separation of mCherry-p62-WT as indicated by the formation of mCherry-p62 clusters (Fig. 7c). With the same concentration and duration of incubation, the results with mCherry-p62-2KR were similar to mCherry-p62-WT, while mCherry-p62-2KQ clearly formed more and bigger clusters (Fig. 7c, d). Finally, we performed fluorescence recovery after photobleaching (FRAP) experiments on mCherry-p62-2KQ puncta and observed the recovery of mCherry signal, which confirmed that the puncta were indeed dynamic liquid droplets (Supplementary Fig. 5). Together, these results suggest that acetylation of p62 facilitates its phase separation with polyubiquitin.

**p62 acetylation promotes the clearance of poly-Ub proteins.** We then checked the role of p62 acetylation in the degradation of intracellular poly-Ub proteins. First, we observed that in *p62*-KO cells and *p62*-KO cells stably expressing p62-2KQ or p62-2KR, starvation normally stimulated the formation of punctate GFP-LC3B structures (Fig. 8a). In p62-2KQ cells, there was extensive colocalization of ubiquitin puncta with GFP-LC3B structures (Fig. 8a, b), whereas in p62-2KR cells, the anti-Ub antibody hardly detected any ubiquitin puncta (Fig. 8a, b). Next, we examined the flux of autophagic degradation of poly-Ub proteins in the cells stably expressing p62-WT, p62-2KQ or p62-2KR upon starvation. We treated the cells with the lysosomal inhibitor bafilomycin A1 (Baf-A1), and detected the presence of poly-Ub proteins. Compared with p62-WT cells, significantly more poly-Ub proteins accumulated in p62-2KQ cells and significantly less in p62-2KR cells (Fig. 8c, d). We also checked the degradation of the p62 proteins themselves in the stable cell lines. Under cell starvation, the turnover of p62-2KQ was clearly much faster than p62-WT and the turnover of p62-2KR was much slower (Fig. 8e, f). Last, we compared the viability of the three cells lines to determine the physiological role of p62 acetylation and p62 body formation upon nutrient deprivation. As expected, under nutrient-deficient conditions with or without Baf-A1, p62-2KQ cells had a higher viability than p62-WT cells, while p62-2KR cells demonstrated a markedly reduced viability (Fig. 8g). These results therefore suggest that both p62 acetylation-mediated p62

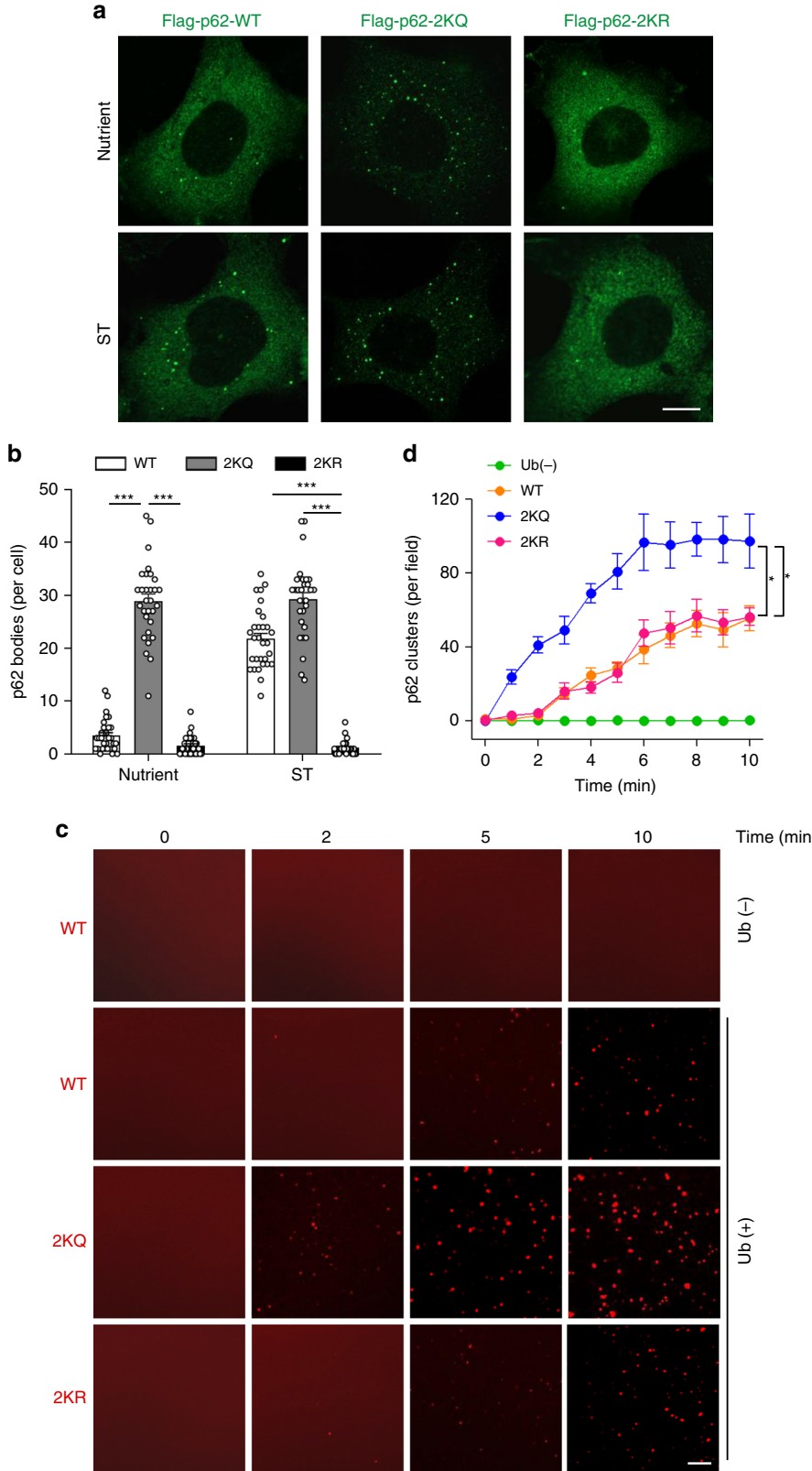

**Fig. 7 Acetylation enhances p62 body formation. a** Acetylation of p62 affects the formation of p62 bodies. *p62*-KO HEK293 cells stably expressing Flag-tagged WT p62, p62-2KQ or p62-2KR were treated with or without starvation, then the cells were stained with anti-Flag and imaged by confocal microscopy. Scale bars, 5 μm. **b** Quantification of p62 puncta in (**a**). The data are presented as mean ± S.E.M., $n = 30$ cells. Two-tailed *t* test, ***$p < 0.001$. **c** In vitro phase separation of p62. Purified recombinant mCherry-tagged WT p62 or each of the indicated p62 mutant proteins was incubated with linear 8 × ubiquitin and visualized by confocal microscopy at different time points. mCherry-p62, 5 μM; linear 8 × ubiquitin, 1.5 μM. Scale bars, 5 μm. **d** Quantification of the number of mCherry-p62 clusters in **c**. The data are presented as mean ± S.E.M., $n = 5$ fields. Two-tailed *t*-test, *$p < 0.05$. Source data are provided as a Source Data file.

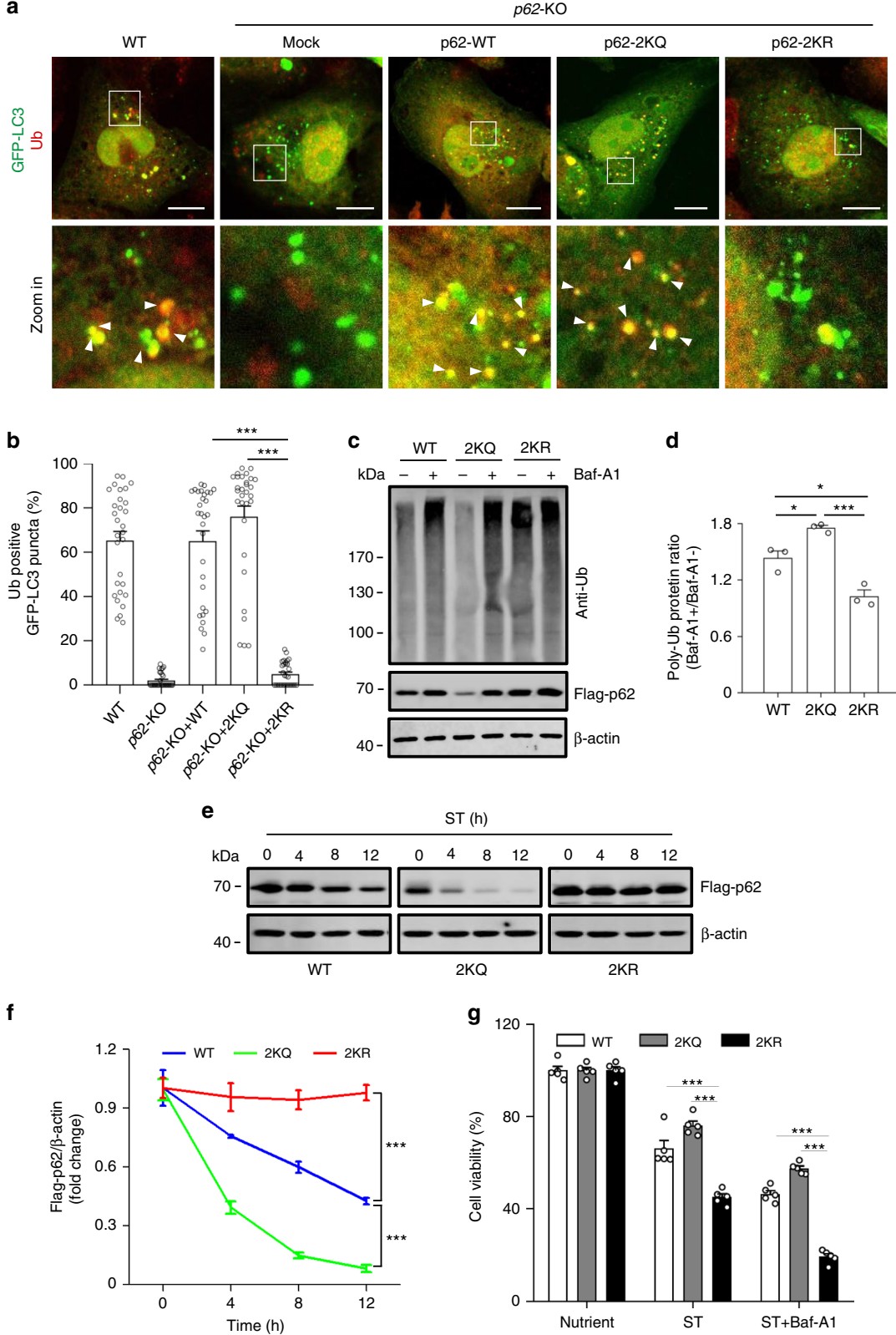

body formation and autophagic degradation of p62 bodies contribute to cell survival under nutrient stress.

## Discussion

We have revealed a previously unknown post-translational modification of p62 which is essential for the function of p62 in

assembling intracellular ubiquitylated proteins under nutrient deficiency. While the data highlight the importance of acetylation in the UBA domain of p62 for cells to cope with nutrient stress, they also suggest that modification of the UBA domain is specifically determined by regulators upstream of p62 that sense and/or respond to the stress, because the phosphorylation and ubiquitylation in the same domain that occur in response to other cell

**Fig. 8 p62 acetylation promotes the clearance of poly-Ub proteins. a** Sequestration of p62 bodies by autophagosomes. *p62*-KO HEK293 cells stably expressing Flag-tagged WT p62, p62-2KQ or p62-2KR were transfected with GFP-LC3B and subjected to starvation. The cells were then stained with anti-Ub and imaged by confocal microscopy. Scale bars, 10 μm. **b** The percentage of GFP-LC3B puncta in **a** that were also positive for ubiquitin. The data are presented as mean ± S.E.M., n = 30 cells. Two-tailed *t*-test, ***p < 0.001. **c**, **d** The level of poly-Ub proteins in *p62*-KO HEK293 cells stably expressing Flag-tagged WT p62 or the p62 mutants. The cells were starved with or without Baf-A1 for 24 h and analyzed by western blot using anti-Ub (**c**). The ratios of the intensity of ubiquitylated proteins in Baf-A1-treated samples over that in Baf-A1-untreated samples are presented in **d**. The data are presented as mean ± S.E.M. of three independent experiments. Two-tailed *t*-test, *p < 0.05, ***p < 0.001. **e**, **f** The degradation of Flag-p62 in *p62*-KO HEK293 cells stably expressing Flag-tagged WT p62 or the p62 mutants upon cell starvation. Flag-p62 was detected by western blot using anti-Flag (**e**), and the quantifications are presented as mean ± S.E.M. of three independent experiments (**f**). Two-tailed *t*-test, ***p < 0.001. **g** Viability of *p62*-KO HEK293 cells stably expressing Flag-tagged WT p62 or the p62 mutants after cell starvation with or without Baf-A1 for 48 h. The data are presented as mean ± S.E.M. of five independent experiments. Two-tailed *t*-test, ***p < 0.001. Source data are provided as a Source Data file.

stresses are not seen upon starvation[20–22]. In addition, the effect of acetylation is achieved by facilitating the phase separation of p62 proteins, which depends on the interaction of the UBA of p62 with polyubiquitin on ubiquitylated proteins.

It is already established that components of the core machinery of autophagy are acetylated. We have now shown that acetylation of autophagy receptor also occurs at least in starved cells. The capture of p62 by autophagosomes relies on both its direct interaction with LC3 on autophagic membranes and the formation of p62 bodies[23,27,34]. We found that acetylation specifically regulates p62 body formation by impacting the affinity of p62 to ubiquitin, and may not affect its direct affinity to LC3. Recent studies have shown that intracellular proteins and organelles are specifically and sequentially degraded by autophagy in starved cells[41–45]. In these cells, ubiquitylated proteins accumulate and aggregate[6,13,46–48], and autophagosome formation and autophagic clearance of long-lived proteins are not inhibited significantly by p62 deletion[17,49]. These findings suggest that the p62 bodies formed in such conditions are mainly employed for selective removal of unwanted cell components, instead of bulk degradation for the production of energy and nutrients. It seems that in response to nutrient deficiency, the activation of acetyltransferase TIP60 plays a key role in coordinating the non-selective and selective processes. TIP60 activation results in the acetylation of ULK1 and ATG3 for the initiation of autophagy[30,32], and the acetylation of p62 for the formation of p62 bodies.

Modifications at many sites in the UBA domain of p62 have been found to affect the association of p62 and ubiquitin[20–22]. Among these sites, K420 is likely one of the key residues because of its location at the interface of two UBAs[18], where it forms a salt bridge with Glutamate 409 (E409) in another UBA to stabilize the intermolecular dimer[18]. It has been shown that K420 ubiquitylation can destabilize UBA dimers[22]. We found that K420 acetylation, a less bulky modification of the lysine side chain, exerts the same effect, possibly by neutralizing the positive charge of K420. Interestingly, our NMR spectroscopy analyses suggest that acetylation at K435, a flexible residue near the C-terminus of UBA, significantly strengthens the K420 acetylation-dependent dimer-to-monomer transition of UBA, which allows the UBA monomer to be more readily available for ubiquitin binding. In addition, our study suggests that K435 acetylation directly promotes the interaction between the UBA of p62 and ubiquitin by alleviating unfavorable electrostatic forces. It should be noted that these conclusions are mainly based on the use of acetylation-mimetic p62 mutants, which may not fully represent acetylated p62 proteins. Nevertheless, our data may provide novel mechanistic insights into the role of acetylation in regulating p62-ubiquitin interaction and the biogenesis of p62 bodies.

In addition to K420 and K435, our mass spectrometry analysis suggested K141 and K189 as two other acetylated residues (Supplementary Fig. 1b). Considering that these residues are located in the region of p62 (amino acids 117–266) which

mediates the interaction with kinase RIP1[50], their acetylation might be involved in the function of p62 as a signal protein.

## Methods

**Cell cultures, reagents, and antibodies.** HeLa (ATCC® CCL-2™), HEK293 (ATCC® CRL-1573™), and HEK293T (ATCC® CRL-11268™) cells were maintained in Dulbecco's modified Eagle medium (Gibco, 11965092) supplemented with 10% fetal bovine serum (Gibco, 10091148) at 37 °C in 5% $CO_2$. *p62*-KO HEK293 cells were generated by transient transfection of pEP-*p62*-KO plasmid followed by selection with puromycin. *p62*-KO HEK293 cells stably expressing Flag-p62-WT, Flag-p62-2KQ, or Flag-p62-2KR were generated by infected with retrovirus which contains corresponding cDNA for 72 h and selected with puromycin.

All chemicals were from Selleck. Bafilomycin A1 (S1413) was used at 2 μM; Puromycin (S7417) was used at 2.5 μg/ml; Trichostain A (S1045) was used at 1 μM for 12 h; Nicotinamide was used at 10 mM for 6 h; Tubacin (S2239) was used at 1 μM for 12 h; MG132 (S2619) was used at 5 μM for 6 h.

The following primary antibodies were used: rabbit polyclonal antibody to ATG7 (Sigma-Aldrich, A2856, 1:1000 for western blot), rabbit monoclonal antibody to acetylated α-tubulin (Lys40) (Sigma-Aldrich, SAB5600134, 1:1000 for western blot), mouse monoclonal antibodies to β-actin (Sigma-Aldrich, A5316, 1:3000 for western blot) and α-tubulin (Sigma-Aldrich, T5293, 1:3000 for western blot), rabbit polyclonal antibodies to Flag (HuaBio, 0912-1, 1:1000 for western blot, 1:100 for immunoprecipitation and immunofluorescence) and HA (HuaBio, 0906-1, 1:1000 for western blot, 1:100 for immunoprecipitation), rabbit polyclonal antibodies to p300 (Santa Cruz, SC-585, 1:200 for western blot) and GST (Santa Cruz, SC-33613, 1:500 for western blot), mouse monoclonal antibodies to Flag (Santa Cruz, SC-807, 1:500 for western blot, 1:100 for immunoprecipitation), HA (Santa Cruz, SC-52592, 1:500 for western blot, 1:100 for immunoprecipitation), acetylated lysine (Santa Cruz, SC-32268, 1:500 for western blot, 1:100 for immunoprecipitation), GCN5 (Santa Cruz, SC-365321, 1:200 for western blot), HDAC6 (Santa Cruz, SC-28386, 1:200 for western blot), and ubiquitin (Santa Cruz, SC-8017, 1:500 for western blot, 1:100 for immunofluorescence), rabbit polyclonal antibody to GFP (MBL, 598, 1:1000 for western blot), rabbit polyclonal antibodies to PCAF (Cell Signaling Technology, C14G9, 1:1000 for western blot), acetylated lysine (Cell Signaling Technology, 9441, 1:1000 for western blot, 1:100 for immunoprecipitation), and phospho-tyrosine (Cell Signaling Technology, 9411, 1:1000 for western blot), rabbit polyclonal antibodies to TIP60 (Proteintech, 10827-1-AP, 1:1000 for western blot) and p62 (Proteintech, 18420-1-AP, 1:1000 for western blot, 1:100 for immunoprecipitation), rabbit polyclonal antibody to phospho-serine (Invitrogen, 61-8100, 1:1000 for western blot). The secondary donkey anti-mouse IRDye680 (LI-COR Biosciences, 926-32222, 1:3000 for western blot) and anti-rabbit IRDye800CW (LI-COR Biosciences, 926-32213, 1:3000 for western blot), donkey anti-rabbit Alexa Fluor 488 (Invitrogen, A-21206, 1:300 for immunofluorescence) and goat anti-mouse Alexa Fluor 635 (Invitrogen, A31574, 1:300 for immunofluorescence).

**Plasmid constructs and transfection.** GFP-LC3B, p300-Flag, and GST-LC3B have been described previously[51,52]. Flag-p62 was made by cloning mouse p62 protein ORF (GeneBank: NM_011018.3) into a pCDNA3-Flag vector using EcoRI and XhoI restriction sites. GST-p62 was generated by cloning mouse p62 protein ORF (GeneBank: NM_011018.3) into a pGEX-4T-1 vector using EcoRI and XhoI restriction sites. mCherry-p62 was generated by cloning p62 protein ORF (GeneBank: NM_011018.3) into a pmCherry-C1 vector at XhoI and EcoRI sites, then cloning the corresponding mCherry-p62 DNA fragment into a pGEX-4T-1 vector at EcoRI and NotI sites to get GST-mCherry-p62. GST-8 × ubiquitin was generated by cloning the corresponding UBC sequence into a pGEX-4T-1 vector at BamHI and NotI restriction sites to get GST-8 × ubiquitin. 6 × His-tagged ubiquitin was generated by cloning the corresponding UBC sequence into a pET-32a vector at XhoI and EcoRI sites. Flag-tagged p62 truncated mutants were generated by cloning the corresponding *p62* DNA fragments into a pCDNA3-Flag vector using EcoRI and XhoI restriction sites. GFP-p62-254-310 and GFP-p62-311-384 was generated by cloning the corresponding *p62* DNA fragments into a pEGFP-C1 vector using

EcoRI and XhoI sites. GST- and 6 × His-tagged p62-UBAs were generated by cloning the corresponding *p62* DNA fragments into a pGEX-4T-1 vector and a pET-32a vector, respectively, using EcoRI and XhoI sites. Site-directed mutagenesis was performed using QuikChange II XL (Stratagene, 210522). pEP-*p62*-KO plasmid was made by cloning the target DNA sequence of human *p62* into a pEP-KO-Z1779 vector using SapI. The primers used in this study were listed in the Supplementary Table 1. Flag-tagged HDAC1, HDAC2, HDAC3, HDAC6, and HDAC10 were gifts from Y Eugene Chin (Institutes of Biology and Medical Science, Soochow University, China). HA-TIP60, HA-PCAF, and HA-GCN5 were gifts from Qiming Sun (School of Medicine, Zhejiang University, China). CBP-Flag was a gift from Jimin Shao (School of Medicine, Zhejiang University, China). Htt-poly97Q-GFP was a gift from William E. Moerner (Department of Chemistry, Stanford University, USA). According to the manufacturer's instructions, indicated plasmids were transfected using Lipofectamine 2000 (Invitrogen, 11668019).

**Cell starvation.** Unless otherwise specified, cells were washed twice with PBS (Sangon Biotech, E607008), followed by addition of EBSS (Gibco, 24010043) and cultured for 1 h.

**RNA interference.** Cells expressing corresponding shRNA were established by using lentivirus-based system. Briefly, scrambled sequence, *p300*, *PCAF*, *GCN5*, *TIP60*, or *HDAC6* targeted sequence was inserted into a lentivirus-based shuttle vector pGLV2 (GenePharma, C06002) and was cotransfected with pRev, pVSV-G, and pGag/Pol into HEK293T cells for 72 h. Then the supernatant was collected and the virus particles were acquired with centrifugation. Unless otherwise specified, cells were infected with the virus for 72 h. The shRNAs used in this study were listed in Supplementary Table 2.

**Immunoprecipitation and western blot.** For p62 ubiquitylation analysis, cells were lysed in urea buffer (50 mM Tris–HCl pH 8.0, 8 M urea, 40 mM imidazole, 100 mM NaH$_2$PO$_4$ and 0.5% CHAPS) added with protease inhibitor cocktail (Roche, 4693159001). Otherwise, cells were lysed in NP-40 buffer (50 mM Tris–HCl pH 7.4, 150 mM NaCl, 1% NP-40 [Sangon Biotech, A100109], 10% glycerol, 2 mM EDTA, 1 mM DTT) supplemented with protease inhibitor cocktail, then the cell lysate was mixed with antibodies at 4 °C for overnight followed by the addition of protein A/G sepharose beads (Thermo Scientific, 20421). Immuno-complexes were washed and subjected to western blot.

For western blot, cells were harvested and lysed in RIPA buffer (50 mM Tris–HCl pH 7.4, 150 mM NaCl, 0.1% SDS, 1% Triton X-100, 1% sodium deoxycholate, 1 mM EDTA) added with protease inhibitor cocktail. Then resolved cell proteins were separated on SDS polyacrylamide gels, and transferred to polyvinylidene difluoride membrane (PALL, bsp0161). After blocking with 5% (w/v) bovine serum albumin (Sigma-Aldrich, B2064), the membrane was stained with the corresponding primary and secondary antibodies. The specific bands were analyzed with an Odyssey® infrared imaging system (Li-Cor Biosciences, Lincoln, NE, USA) and quantified using Image J. All uncropped blots/gels are shown in Supplementary Figs. 6–12.

**In vitro acetylation assay.** GST-p62-D69A purified from *E. coli* and HA-tagged WT TIP60 or TIP60-DN (Q377E, G380E) purified from HEK293T cells were incubated in HAT assay buffer (50 mM Tris–HCl pH 8.0, 10% glycerol, 0.1 mM EDTA, 1 mM dithiothreitol) with acetyl-coenzyme A at 30 °C for 3 h. Then the reaction products were analyzed by western blot.

**In vitro α-tubulin deacetylase assay.** Porcine brain-derived microtubules (Cytoskeleton, Inc, ML116) were incubated with Flag-tagged WT HDAC6 or HDAC6-DN (H216A, H611A) purified from HEK293T cells in the deacetylase assay buffer (10 mM Tris–HCl pH 8.0, 10 mM NaCl) at 30 °C for 0.5 h. Then the reaction mixtures were analyzed by western blot.

**Protein expression and purification.** GST-tagged LC3B, p62-D69A, mCherry-p62, and mCherry-p62 mutants, p62-UBA and p62-UBA mutants, and 8 × ubiquitin were expressed in *E. coli* BL21 cells (Transgen Biotech, CD601) by induction with of 0.5 mM isopropyl β-D-thiogalactopyranoside/IPTG (Sigma-Aldrich, PHG0010) at 18 °C for 24 h. The proteins were purified by using glutathione-sepharose 4B beads (GE Healthcare Life Sciences, 17-0756-01), then the proteins were eluted with glutathione (Beyotime, S0073) or incubated with TEV protease (a gift from Qiming Sun, Zhejiang University, Hangzhou, China) at 4 °C for 4 h to release the proteins from the GST. Last, the protein eluate was concentrated with Amicon Ultra-4 filter (Millipore, UFC801024) and glycerol was added to a final concentration of 25% for storage at −80 °C.

The proteins used for NMR spectra were expressed as fusion proteins with 6 × His tag in *E. coli* BL21 cells. For isotope enrichment, the cells were grown in M9-minimum medium, and the $^{15}$NH$_4$Cl (Sigma-Aldrich, 299251) was supplied as the sole nitrogen source. The proteins were purified by using Ni:NTA (GE Healthcare Life Sciences, 17-5255-01) column chromatography, the N-terminal 6 × His tag was removed from the proteins by using thrombin protease (Sigma-Aldrich, T4648).

The proteins were further purified by SourceQ anion-exchange column (GE Healthcare Life Sciences, 17-0947-01). Protein purity and identity were confirmed by SDS-PAGE and ESI-MS.

**In vitro GST pull-down assay.** For protein pull-down assay, purified GST, GST-p62-D69A, GST-p62-UBA, GST-p62-UBA mutant or GST-LC3B protein was incubated with indicated purified proteins or cell lysates in NP-40 buffer at 4 °C for 4 h. Then glutathione-sepharose 4B beads were added to the mixture followed by incubation at 4 °C for 2 h. The beads were washed and used for western blot.

**Mass spectrometry analysis.** The samples were precipitated and resolved by 8 M urea, and then treated with 5 mM TCEP and 10 mM IAA to reduce the disulfide bonds and alkylate the resulting thiol groups, sequentially. The mixture was digested for 16 h at 37 °C by trypsin at an enzyme-to-substrate ratio of 1:50 (wt/wt). The trypsin-digested peptides were loaded on an in-house packed capillary reverse-phase C18 column (15 cm length, 100-μm i.d. × 360-μm o.d. 3 μM particle size, 100 Å pore diameter) connected to a Thermo Easy-nLC1000 HPLC system. The samples were analyzed with a 90 min-HPLC gradient from 0 to 100% of buffer B (buffer A: 0.1% formic acid in water; buffer B: 0.1% formic acid in acetonitrile) at 300 nL/min: 0–1 min, 0–4% B; 1–76 min, 4–35% B; 76–84 min, 35–60% B; 84–85 min, 60–100% B; 85–90 min, 100% B. The eluted peptides were ionized and directly introduced into a Q-Exactive mass spectrometer using a nano-spray source with a distal 1.8-kV spray voltage. Survey full-scan MS spectra (from *m/z* 300 to 1800) was acquired in the Orbitrap analyzer with resolution *r* = 70,000 at *m/z* 200. One acquisition cycle includes one full-scan MS spectrum followed by top 20 MS/MS events, sequentially generated on the first to the twentieth most intense ions selected from the full MS spectrum at a 28% normalized collision energy. The acquired MS/MS data were analyzed based on UniProtKB Human database (released on Sept. 19, 2016) or UniProtKB *E. coli* database (released on Nov. 11, 2016) containing SQSTM1 (Q64337) using Integrated Proteomics Pipeline (IP2, http://integratedproteomics.com/). In order to accurately estimate peptide probabilities and false discovery rates (FDR), we used a decoy database containing the reversed sequences of all the proteins appended to the target database. FDR was set at 0.01. Mass tolerance for precursor ions was set at 50 ppm. Trypsin was defined as cleavage enzyme and the maximal number of missed cleavage sites was set at 3. Carbamidomethylation (+57.02146) of cysteine was considered as a static modification. Lysine acetylation and methionine oxidation were set as variable modifications. The modified peptides were manually checked and labeled.

**Nuclear magnetic resonance (NMR) collection.** NMR experiments were performed on a 600 MHz instrument equipped with cryogenic probes (Bruker Biospin, Billerica, MA) at 298 K in pH 6.0 20 mM sodium acetate buffer containing 100 mM NaCl.

For the dilution experiment of the p62-UBA, the $^{15}$N-labeled p62-UBA was sequentially diluted from 500 to 20 μM. For the ubiquitin titration assay, unlabeled ubiquitin was added into the $^{15}$N-labeled 200 μM p62-UBA protein solution. All NMR data were analyzed by using NMRPipe (version 2010.160.15.01)[53] and CcpNmr (version 2.4.2)[54]. The normalized chemical shift perturbation was calculated using the equation $[0.5 × (\Delta\delta H^2 + 0.2 × \Delta\delta N^2)]^{0.5}$, in which $\Delta\delta H$ and $\Delta\delta N$ are the CSP in ppm unit in the proton and nitrogen dimensions, respectively.

For the p62-UBA dimer dissociation process, it can be described in Eq. (1).

$$D \overset{K_{D,dimer}}{\rightleftharpoons} M + M \qquad (1)$$

in which D and M represent p62-UBA dimer and monomer, respectively.

The equilibrium dissociation constant of p62-UBA dimer can be by Eq. (2).

$$K_{D,dimer} = \frac{[M]^2}{[D]} \qquad (2)$$

in which [M] and [D] are the concentration of monomer and dimer, respectively. The relationship between [M] and [D] and the total amount of the protein $[M]_T$ can be described by Eqs. (3) and (4).

$$[M] = [M]_T * V_m / (V_m + V_d) \qquad (3)$$

$$[D] = [M]_T * V_d / [2 * (V_m + V_d)] \qquad (4)$$

in which $V_m$ and $V_d$ represent the volume of the peak corresponding monomer and dimer in HSQC spectrum, respectively. The equilibrium dissociation constant of p62-UBA dimer $K_{D,dimer}$ in Fig. 6c was calculated by substituting Eqs. (3) and (4) into Eq. (2).

The ubiquitin titration of p62-UBA process can be described by Eq. (5).

$$D \overset{1}{\underset{K_{D,dimer}}{\rightleftharpoons}} M + M \overset{2}{\underset{\substack{K_{D,complex} \\ +ubiquitin}}{\rightleftharpoons}} complex \qquad (5)$$

The total amount of the p62-UBA $[M]_T$ in process dissociation event (1) of Eq. (5) can be expressed as Eq. (6).

$$[M]_T = [M] + [D] * 2 \qquad (6)$$

This equation can be rearranged as Eq. (7).

$$[D] = \frac{[M]_T - [M]}{2} \quad (7)$$

Substituting Eq. (7) into Eq. (2) affords Eq. (8).

$$K_{D,dimer} = \frac{2^*[M]^2}{[M]_T - [M]} \quad (8)$$

Solving the Eq. (8) yields Eq. (9).

$$[M] = \frac{-K_{D,dimer} + \sqrt{K_{D,dimer}^2 + 8^*[M]_T^* K_{D,dimer}}}{4} \quad (9)$$

Regarding the binding event (2) in Eq. (5), we can further obtain Eq. (10), as described previously[55].

$$\Delta\delta_{obs} = \Delta\delta_{max}\{([M]+[P]+K_{D,complex}) - \sqrt{([M]+[P]+K_{D,complex})^2 - 4^*[M]^*[P]}\}/(2^*[M]) \quad (10)$$

Here, $\Delta\delta_{obs}$ is the observed chemical shift perturbation, $\Delta\delta_{max}$ is the chemical shift perturbation at the end of the titration, $K_{D,complex}$ is the equilibrium dissociation constant between p62-UBA monomer and ubiquitin, [P] is the concentration of ubiquitin, and [M] is the concentration of p62-UBA monomer, which can be calculated from Eq. (9). The value of $K_{D,complex}$ can be calculated by fitting $\Delta\delta_{obs}$ against [P] using Eq. (10).

**Molecular dynamics (MD) simulations**. The MD simulations were performed using AMBER 16 package with ff14SB force field[56,57]. The crystal structure of p62-UBA (PDB: 3B0F) was used as the starting structure. Three different types of acetylation model (K420 or/and K435 acetylation) and WT structure were used for the MD simulation. The force field parameters of acetylated lysine were described previously[58]. The protein conformation was solvated in a TIP3P water box with 10 Å in every direction. The $Na^+$ ions were added to neutralize the charge of the whole system. Non-bonded interaction cutoff was set to 10 Å. The enhanced accelerated molecular dynamics were performed to sampling the conformation change of p62-UBA dimer more efficiently[59]. The whole MD production process was lasts for 400 ns to produce 4000 snapshots at 100 ps interval. The RMSD analysis was performed using CPPTRAJ module in AMBER 16.

**Structure model of p62-UBA-ubiquitin**. The structural model of p62-UBA and ubiquitin complex was acquired by using PyMOL and the electrostatic surface of ubiquitin was generated by using APBS plugin in PyMOL.

**Paramagnetic relaxation enhancement (PRE) NMR measurement**. PRE experiment was performed as described previously[55]. Briefly, the ubiquitin E24C mutant was reacted with 10-fold excess of 4-(2-Iodoacetamido)-2,2,6,6-tetra-methylpiperidine 1-Oxyl Free Radical (TMPO) overnight at room temperature (RT). Unreacted probe was removed by desalting. Protein purity and identity were confirmed by SDS-PAGE and ESI-MS. Transverse relaxation rates of amide protons for the $^{15}N$-labeled p62-UBA were measured using the standard pulse scheme with a 12 ms delay between the two time points. To make sure the $^{15}N$-labeled p62-UBA was 100% bound, the nitroxide probe labeled ubiquitin was 6-fold excess.

**Immunostaining and cofocal microscopy**. For immunostaining, HEK293 cells were fixed in 4% formaldehyde for 10 min at RT followed by permeabilization and blocking with PBS containing 10% fetal calf serum (Sigma-Aldrich, F0685) and 0.1% saponin (Sigma-Aldrich, S7900) for 30 min. Then the cells were incubated with corresponding primary and secondary antibodies. The images were obtained on the confocal microscope (LSM800 with Airyscan, Carl Zeiss, Germany). Axiovision Automatic measurement program was used to counting the number of p62 puncta and LC3B puncta. p62 puncta with diameter >0.2 μm were scored as positive; LC3B puncta with diameter between 0.2 and 1 μm were scored as positive.

**p62 phase separation**. Purified 5 μM mCherry-p62 protein and 1.5 μM 8 × ubiquitin protein were incubated in phase separation buffer (40 mM Tris–HCl pH 7.4, 150 mM NaCl, 10% glycerol, 1 mM DTT) on the chambered cover glass (Thermo Scientific, 155411) which was previously coated with 1 mg/ml BSA for the indicated time at RT, then the reaction was imaged by confocal microscope.

**Fluorescence recovery after photobleaching**. p62 clusters were formed as described above for 10 min in the cover glass at RT. The clusters were bleached with 561 nm laser and imaged with a LSM800 confocal microscope (Carl Zeiss).

**Cell viability assay**. HEK293 cells were seeded in 96-well plates ($3.0 \times 10^3$ cells per well) and incubated 24 h before being treated as indicated for 48 h. Cell viability was measured by adding 10% CCK-8 (Bimake, B34304) for an additional 1 h. Absorbance was measured at 450 nm. The cell viability of wells containing untreated cells and wells containing medium only were set as 100% and 0%, respectively, all other viabilities were normalized to these values.

**Statistical analysis**. All the statistical data are presented as mean ± S.E.M., statistical significance of the differences was determined using the Student $t$ test. $p < 0.05$ was considered statistically significant.

**Reporting summary**. Further information on research design is available in the Nature Research Reporting Summary linked to this article.

## Data availability

The original data were deposited to Mendeley Data (https://doi.org/10.17632/4f6vgkrn75.1). The source data underlying Figs. 1, 2, 3a–d, f–h, 4, 5, 6 b, c, e, 7b, d, 8b–g, Table 1, Supplementary Figs. 2, 3b and 4b are provided as a Source Data file. The data that support this study are available from the corresponding authors upon reasonable request.

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

## Acknowledgements

We are grateful to the Imaging Center of Zhejiang University School of Medicine for their assistance in confocal microscopy. We thank Y. Eugene Chin, Qiming Sun, Jimin Shao, and William E. Moerner for the sharing of plasmids and reagents. This study was supported by the National Natural Science Foundation of China (31790402), the National Basic Research Program of China (2017YFA0503402), and the National Natural Science Foundation of China (31530040, 31671434).

## Author contributions

Z.Y.Y., C.T. and W.L. designed the experiments. Z.Y.Y., W.X.J., L.Y.Q, Z.G., W.W., J.L., Y.S.W. and H.T.Z. performed the experiments. C.P. performed the mass spectrometry. Z.Y.Y., T.H.Z., C.T. and W.L. wrote the paper. All authors discussed the results and commented on the paper.

## Competing interests

The authors declare no competing interests.
