## [Peer Review File · Nature Communications]

Reviewers' comments:

Reviewer #1 (Remarks to the Author):

The manuscript by You et al reports that during starvation the autophagy cargo receptor p62 is acetylated at two lysine residues within its UBA domain. Acetylation and deacetylation on these sites are mediated by the TIP60 and HDAC6 proteins, respectively. The acetylation interferes with the autoinhibitory dimerization of the UBA domain and thereby enhances ubiquitin binding. In addition, acetylation on one of the two lysines increases the affinity of the UBA domain for ubiquitin. Furthermore, it is shown that an acetylation-mimicking mutant of p62 shows an enhanced propensity to phase separate in the presence of ubiquitin and that acetylation of p62 is required for the efficient degradation of ubiquitinated proteins in the lysosome upon starvation.

This is an extensive, well written study. The data are clean and the message is interesting to a wider community working on autophagy, proteins quality control and degradation as well as protein acetylation. In my opinion the manuscript is worthy of publication in this journal, pending a few points the authors should address.

Major points:

1. The authors should use a catalytic mutant of TIP60 in the experiment shown in Fig. 3d. In addition, the authors should do an experiment analogous to the one shown with HDAC6-DN in Fig. 4c. I.e. they should test if an acetylation deficient mutant of TIP60 can complement shTIP60-1 treated cells (Fig. 3c).
2. The authors should explain why they used different cell lines for their experiments. For example, why was the experiment shown in Fig. 3f conducted with HeLa cells, while most of the experiments were conducted with HEK293T cells? The cell lines used should be clearly stated in the legends.
3. It is not clear to this reviewer why there is a difference in the cell viability between the WT, 2KQ and 2KR mutants after Baf-A1 treatment (Fig 8g). If the beneficial effect of acetylation is the promotion of ubiquitinated proteins by autophagy it should at least be reduced upon the block of their degradation by BAF-A1. Can the authors please elaborate on this aspect?

4. Page 5: The authors should state clearly in the results section and the legend, which phosphorylation site they detected in Fig. 1c.

Minor points:

5. The authors refer to p62 as adaptor protein (page 3). However, in the field this class of molecules is commonly referred to as “cargo receptors”.

6. Page 3, lines 18: The authors write that p62 forms dimmers (should be dimers) via its UBA domain. However, this is misleading as p62 is an oligomer (see for example PMIDs: 26413874 and 25921531). This sentence should rather say that the UBA of p62 dimerizes.

7. The authors write on page 5 that the K7A and K7A/D69A mutants would abolish oligomerization. However, they only reduce oligomerization and the resulting mutants are not monomers. This statement should therefore be toned down.

8. page 5: RESULTS should be RESULTS

Reviewer #2 (Remarks to the Author):

This is a well written manuscript that examines the post translational modification of p62 through acetylation and its role in functional regulation and protein aggregation.

The system has been well studied structurally in the literature by both NMR and x-ray crystallography and the methodologies for studying the monomer-dimer equilibrium for the p62 UBA domain have been reported in earlier studies. The authors use this biophysical approach to characterise the impact of the K420 and K435 acetylation on these equilibria, including the affinity for the ubiquitin monomer.

The investigation is thorough and quantitative and provides clear structural insights in to the effects on the stability of the UBA dimer. A figure to show the localisation of the side chains close to the dimer interface would be helpful. The MD simulations provide an interesting insight into the possible

steric effects from acetylation that destabilise the dimer, along with the important control experiments to justify this approach.

This aspect of the paper is clearly presented and adds considerable structural insight as to the molecular basis of the in vivo observations.

Reviewer #3 (Remarks to the Author):

In this manuscript You et al identify a novel regulatory mechanism controlling degradation of ubiquitinated proteins in conditions of nutrient starvation. The mechanism is mediated by the acetylation of p62 which promotes its interaction with ubiquitinated proteins thus targeting them for autophagic turnover. This is an exciting finding as it significantly contributes to our understanding of how cells maintain their viability during periods of nutrient deprivation. The data is of high quality and provides a convincing evidence for the conclusions made by the authors. All the necessary controls have been performed thoroughly, quantifications have been done where required, the text is well written, structured, logical and easy to read. This reviewer has no reservation in recommending it for publication provided that the following minor points are addressed:

1. The authors write in the abstract and the discussion that acetylation of p62 does not affect its association with autophagic membranes however no evidence is presented in support of this statement. Please show the data that would demonstrate this.
2. In the same line there appears to be a discrepancy between this apparent lack of effect of acetylation on p62 binding to autophagosomes with the data in Fig. 8e,f which shows that Lys mutations affect the turnover of p62. Can the authors provide an explanation for how p62 turnover is affected if it binds to autophagic structures independent of its acetylation state?
3. The conclusion that acetylation disrupts the dimer formation is based on experiments with p62 mutants whilst the effect of acetylation is only assessed by molecular dynamics simulations. Whilst this reviewer recognises that the generation of specifically acetylated recombinant proteins is extremely difficult the authors should acknowledge this as a limitation of their study and tone down their conclusions.
4. Please indicate the n numbers for all figures where quantifications have not been provided.
5. Typos: 'dimmers' on page 3, 'much more effective at pulling down much more' on page 9.

Viktor Korolchuk

Reviewer #1 (Remarks to the Author):

The manuscript by You et al reports that during starvation the autophagy cargo receptor p62 is acetylated at two lysine residues within its UBA domain. Acetylation and deacetylation on these sites are mediated by the TIP60 and HDAC6 proteins, respectively. The acetylation interferes with the autoinhibitory dimerization of the UBA domain and thereby enhances ubiquitin binding. In addition, acetylation on one of the two lysines increases the affinity of the UBA domain for ubiquitin. Furthermore, it is shown that an acetylation-mimicking mutant of p62 shows an enhanced propensity to phase separate in the presence of ubiquitin and that acetylation of p62 is required for the efficient degradation of ubiquitinated proteins in the lysosome upon starvation.

This is an extensive, well written study. The data are clean and the message is interesting to a wider community working on autophagy, proteins quality control and degradation as well as protein acetylation. In my opinion the manuscript is worthy of publication in this journal, pending a few points the authors should address.

Major points:

1. The authors should use a catalytic mutant of TIP60 in the experiment shown in Fig. 3d. In addition, the authors should do an experiment analogous to the one shown with HDAC6-DN in Fig. 4c. I.e. they should test if an acetylation deficient mutant of TIP60 can complement shTIP60-1 treated cells (Fig. 3c).

RE: According to your suggestion, we have created an acetyltransferase-deficient TIP60 mutant (TIP60-DN) in which two residues (Gln-377 and Gly-380) essential for the binding of acetyl CoA were replaced by Glu (PMID: 10966108). We confirmed that this TIP60 mutant is unable to acetylate p62 *in vitro*. The re-expression of WT-TIP60 but not the acetyltransferase deficient TIP60 rescued the reduction of p62 acetylation caused by TIP60 KD. These data have been shown in Figs. 3c and 3d respectively in the revised manuscript.

2. The authors should explain why they used different cell lines for their experiments. For example, why was the experiment shown in Fig. 3f conducted with HeLa cells, while most of the experiments were conducted with HEK293T cells? The cell lines used should be clearly stated in the legends.

RE: In fact, most of the experiments were performed in both HeLa cells and HEK293 cells and consistent results were obtained. HeLa cells have a relative higher basal level of p62

acetylation compared to HEK293 cells. To better show the decrease in the acetylation by treatments, we chose to present the data from HeLa cells, when we presented the data from HEK293 cells to better show the increase in the acetylation by treatments. According to your suggestion, in each figure legend, we have clearly stated the cell lines used in the experiments.

3. It is not clear to this reviewer why there is a difference in the cell viability between the WT, 2KQ and 2KR mutants after Baf-A1 treatment (Fig 8g). If the beneficial effect of acetylation is the promotion of ubiquitinated proteins by autophagy it should at least be reduced upon the block of their degradation by BAF-A1. Can the authors please elaborate on this aspect?

RE: Our explanation is that formation of p62 body itself can attenuate the cytotoxicity of misfolded/unfolded proteins although p62 bodies are still toxic to cells. Compared to p62-WT and p62-2KR, p62-2KQ is able to assemble ubiquitylated proteins to form more p62 bodies, therefore even these p62 bodies cannot be degraded in cells under starvation and Baf-A1 treatment, p62-2KQ cells still have the highest survival rate. Our data support previous studies showing that cellular inclusion body formation can weaken the toxicity of abnormal/misfolded huntingtin or α -synuclein proteins (PMID:15483602, 14627698). The increased survival difference between p62-2KR and p62-2KQ cells under starvation and Baf-A1 treatment may reflect the increased sensitivity to misfolded/unfolded proteins of these cells, because compared to starvation only, starvation and Baf-A1 treatment can aggravate nutrient deficiency due to the blockage of non-selective autophagy.

4. Page 5: The authors should state clearly in the results section and the legend, which phosphorylation site they detected in Fig. 1c.

RE: Thank you. A description about the antibodies has been added in the results and the legend. In fact, in the experiments of Fig. 1c, we used two panphosphorylation antibodies against phospho-serine and phospho-tyrosine respectively. We know that it may be the best to use the site-specific antibodies against p-p62-S403 (phosphorylated by CK2 and TBK1) and p-p62-S407 (phosphorylated by ULK1), but they were not commercially available at the time we performed the experiments and previous study (PMID: 25723488) has shown that p62 is not phosphorylated at these sites under nutrient limitation.

Minor points:

5. The authors refer to p62 as adaptor protein (page 3). However, in the field this class of molecules is commonly referred to as “cargo receptors”.

RE: Thanks. We have changed it to “cargo receptor” in the manuscript.

6. Page 3, lines 18: The authors write that p62 forms dimmers (should be dimers) via its UBA domain. However, this is misleading as p62 is an oligomer (see for example PMIDs: 26413874 and 25921531). This sentence should rather say that the UBA of p62 dimerizes.

RE: We have corrected them according to your kind suggestion.

7. The authors write on page 5 that the K7A and K7A/D69A mutants would abolish oligomerization. However, they only reduce oligomerization and the resulting mutants are not monomers. This statement should therefore be toned down.

RE: Thanks for pointing out the inaccuracy. We have toned down the description by changing it to “the multimerization-impaired mutants p62-K7A and p62-K7A/D69A” and deleting the word “monomeric”. We also added references to two related papers.

8. page 5: RESULTS should be RESULTS

RE: It has been corrected, thank you.

Reviewer #2 (Remarks to the Author):

This is a well written manuscript that examines the post translational modification of p62 through acetylation and its role in functional regulation and protein aggregation.

The system has been well studied structurally in the literature by both NMR and x-ray crystallography and the methodologies for studying the monomer-dimer equilibrium for the p62 UBA domain have been reported in earlier studies. The authors use this biophysical approach to characterise the impact of the K420 and K435 acetylation on these equilibria, including the affinity for the ubiquitin monomer.

The investigation is thorough and quantitative and provides clear structural insights in to the effects on the stability of the UBA dimer. A figure to show the localisation of the side chains close to the dimer interface would be helpful. The MD simulations provide an interesting insight into the possible steric effects from acetylation that destabilise the dimer, along with the important control experiments to justify this approach.

This aspect of the paper is clearly presented and adds considerable structural insight as to the

molecular basis of the in vivo observations.

RE: We sincerely thank you for your positive comments on our manuscript.

Reviewer #3 (Remarks to the Author):

In this manuscript You et al identify a novel regulatory mechanism controlling degradation of ubiquitinated proteins in conditions of nutrient starvation. The mechanism is mediated by the acetylation of p62 which promotes its interaction with ubiquitinated proteins thus targeting them for autophagic turnover. This is an exciting finding as it significantly contributes to our understanding of how cells maintain their viability during periods of nutrient deprivation. The data is of high quality and provides a convincing evidence for the conclusions made by the authors. All the necessary controls have been performed thoroughly, quantifications have been done where required, the text is well written, structured, logical and easy to read. This reviewer has no reservation in recommending it for publication provided that the following minor points are addressed:

1. The authors write in the abstract and the discussion that acetylation of p62 does not affect its association with autophagic membranes however no evidence is presented in support of this statement. Please show the data that would demonstrate this.

RE: Thank you for your critical comments on this point. It is known that the capture of p62 by autophagosomes relies on both the direct interaction of p62 and LC3 and the formation of p62 bodies (PMID: 21220505, 26413874 and 29507397). What we wanted to say here is that while the acetylation affects the association of p62 with ubiquitin, it may not affect the direct binding affinity between p62 and LC3 on autophagosomes, based on our data showing in Fig. 5d. We agree with you that we don't have direct and sufficient evidence supporting the statement that acetylation of p62 does not affect its association with autophagic membranes. According to your criticism, therefore we deleted the relevant sentence from the Abstract and modified (toned down) the related description in the Discussion in our revised manuscript.

2. In the same line there appears to be a discrepancy between this apparent lack of effect of acetylation on p62 binding to autophagosomes with the data in Fig. 8e,f which shows that Lys mutations affect the turnover of p62. Can the authors provide an explanation for how p62 turnover

is affected if it binds to autophagic structures independent of its acetylation state?

RE: As we explained above, we do think that acetylation of p62 affects the capture of p62 by autophagosomes which is mainly due to its effect on p62 body formation. We admit that our description on this point is inaccurate. According to your criticism, we have deleted the relevant sentence from the Abstract and have modified the description in the Discussion in the revised manuscript.

3. The conclusion that acetylation disrupts the dimer formation is based on experiments with p62 mutants whilst the effect of acetylation is only assessed by molecular dynamics simulations. Whilst this reviewer recognizes that the generation of specifically acetylated recombinant proteins is extremely difficult the authors should acknowledge this as a limitation of their study and tone down their conclusions.

RE: Thank you for your kind suggestion. We indeed spent half a year trying to get the recombinant acetylated p62. Unfortunately, the system didn't work well. According to your suggestion, we have acknowledged the limitation of our experiments using p62 mutants for determining the effect of acetylation on p62-UBA dimer formation. We also toned down the descriptions about our conclusions in the revised manuscript.

4. Please indicate the n numbers for all figures where quantifications have not been provided.

RE: We have indicated the n number for all figures where quantifications have not been provided in Reporting Summary.

5. Typos: 'dimers' on page 3, 'much more effective at pulling down much more' on page 9.

RE: Thank you. We have corrected these errors in our revised manuscript.

REVIEWERS' COMMENTS:

Reviewer #1 (Remarks to the Author):

The authors have addressed all my comments and I have no further points. This is a very strong manuscript.